# Transcriptome Analysis of the Effects of Fasting Caecotrophy on Hepatic Lipid Metabolism in New Zealand Rabbits

**DOI:** 10.3390/ani9090648

**Published:** 2019-09-03

**Authors:** Yadong Wang, Huifen Xu, Guirong Sun, Mingming Xue, Shuaijie Sun, Tao Huang, Jianshe Zhou, Juan J. Loor, Ming Li

**Affiliations:** 1College of Animal Science and Veterinary Medicine, Henan Agricultural University, Zhengzhou 450046, China (Y.W.) (H.X.) (G.S.) (M.X.) (S.S.) (T.H.) (J.Z.); 2Mammalian NutriPhysioGenomics, Department of Animal Sciences and Division of Nutritional Sciences, University of Illinois, Champaign, IL 61801, USA

**Keywords:** fasting caecotrophy, lipid metabolism, rabbits, transcriptome sequencing

## Abstract

**Simple Summary:**

Caecotrophy in small herbivores, including rabbits, is the instinctive behavior of eating soft feces. Little is known about the impact of caecotrophy on growth and metabolism. In the present study, we used an Elizabeth circle to prevent rabbits from eating soft feces and measured changes in feed intake, body weight, internal organ weight, serum biochemical indices and liver lipid droplet accumulation. Liver tissue was also used for transcriptome sequencing. Results indicated that fasting caecotrophy decreased rabbit growth and lipid synthesis in the liver.

**Abstract:**

In order to investigate the effects of fasting caecotrophy on hepatic lipid metabolism in rabbits, 12 weaned female New Zealand white rabbits were randomly divided into (n = 6/group) a control and fasting caecotrophy group. Rabbits in the experimental group were treated with an Elizabeth circle to prevent them from eating their own soft feces for a 60-day period. Growth and blood biochemical indices, transcriptome sequencing and histology analysis of the liver were performed. Compared with the control group, final weight, weight gain, liver weight, growth rate and feed conversion ratio, all decreased in the experimental group (*p* < 0.05). RNA sequencing (RNA-seq) analysis revealed a total of 301.2 million raw reads (approximately 45.06 Gb of high-quality clean data) that were mapped to the rabbit genome. After a five-step filtering process, 14,964 genes were identified, including 444 differentially expressed genes (*p* < 0.05, foldchange ≥ 1). A number of differently expressed genes linked to lipid metabolism were further analyzed including *CYP7A1*, *SREBP*, *ABCA1*, *GPAM*, *CYP3A1*, *RBP4 and RDH5*. The KEGG (Kyoto Encyclopedia of Genes and Genomes) annotation of the differentially expressed genes indicated that main pathways affected were pentose and glucuronide interactions, starch and sucrose metabolism, retinol metabolism and *PPAR* signaling. Overall, the present study revealed that preventing caecotrophy reduced growth and altered lipid metabolism, both of which will help guide the development of new approaches for rabbits’ feeding and production. These data also provide a reference for studying the effects of soft feces in other small herbivores.

## 1. Introduction

Many small herbivores have a natural instinct for performing caecotrophy [1]. Because the small body shape and digestive tract volume is limited, the average residence time of food in the digestive tract is relatively short [2,3]. In order to meet their nutritional needs, small herbivores need to obtain adequate high-quality food primarily from low-quality, highly-fibrous plant stems and leaves [4]. In turn, symbiotic microorganisms in the hind-gut aid the animal in digesting the fiber components [5,6]. Because microbial fermentation takes longer than the average residence time of food in the digestive tract, increasing digestibility by ingesting incompletely-digested nutrients is an important nutritional strategy for small herbivores [7,8,9]. 

There are two types of feces excreted by rabbits: hard feces (nutrient-poor) and soft feces (consisting of protein, vitamins, and inorganic salts) [10,11,12]. The latter contain a large number of microorganisms, which are important for microbial fermentation. Thus, we speculated that rabbits might build their intestinal microbial flora through caecotrophy behavior [13,14]. Although modern feeding and management techniques are designed to fully meet the nutritional needs for growth, rabbits still maintain the habit of eating soft feces [15,16,17]. One study comparing high and low body weight Rex rabbits revealed that the high-weight group had more abundant bacterial groups, with soft feces inducing greater numbers than hard feces [18,19,20]. Intestinal flora and their host have a mutually beneficial symbiosis, one in which metabolism, immune system development, disease resistance and other physiological functions in the host are closely associated with the proper functioning of the gut flora [21,22]. Our team also uncovered that fasting caecotrophy in rabbits resulted in changes in intestinal microbial flora associated with lipid metabolism, suggesting an important role of caecotrophy in this process (unpublished data).

Besides its role in supporting digestion, absorption, decomposition, synthesis and transport of lipids, the liver plays an important role in overall metabolism of lipids [23,24]. Bile acids produced by the liver are important for emulsifying dietary lipids and promoting the digestion and absorption of dietary lipids [25]. By generating ATP for its own utilization, liver is also the main site for the oxidative metabolism of fatty acids and production of ketone bodies [26,27,28]. In monogastrics, liver is the primary site for fatty acid oxidation and ketone body production [29,30]. Liver-synthesized cholesterol accounts for more than 80% of the total cholesterol produced by the body and is the main source of plasma cholesterol [31].

Transcriptome sequencing is an essential tool for the study of large-scale gene expression profiles, particularly in temporal studies or studies involving different treatment conditions [32]. Because of the high accuracy, high throughput, high sensitivity and low cost, this tool has been widely used in research with livestock [33]. Transcriptome sequencing can help us better understand the functional outcome of changes in gene expression in key metabolic organs such as the liver [34]. 

Previous research indicated that caecrophagy decreased the body weight of Rex rabbits [35]. However, whether caecotrophy affects growth performance and lipid metabolism in New Zealand white rabbits is unknown. In order to investigate the effects of fasting caecotrophy on lipid metabolism and the possible mechanisms, we performed a feeding experiment and subjected liver tissue to transcriptome sequencing. The main aim was to clarify the effects of fasting caecotrophy on growth rate and lipid metabolism, hence, providing experimental evidence that could potentially be used to develop novel approaches for the feeding management of rabbits.

## 2. Materials and Methods 

### 2.1. Animals and Experimental Design

#### 2.1.1. Ethics Statement

The present study was designed and performed according to the guidelines of the institutional Animal Care and Use Committee College of Animal Husbandry and Veterinary Medicine of Henan Agricultural University. All experiments involving animals were performed in accordance with protocols and guidelines approved by the Institutional Animal Care and Use Committee (IACUC) of Henan Agricultural University, China (Permit Number: 11-0085; Date: 06-2011).

#### 2.1.2. Animals

Forty-eight healthy female 65-day-old Angora long-haired rabbits with similar body weights (1.28 ± 0.12 kg) from a commercial farm (Jiaozuo, Henan, China) were selected and raised in the animal experimental center of Henan Agriculture University. These rabbits were randomly divided into three groups (16 replicates in each group, 1 rabbit for each replicate): group A was the control group and fed according to normal procedures (CON group); group B was the coprophagy allowed group, in which rabbits wore a narrow collar (2.5 cm) that did not prevent them from eating soft feces (CA group, Figure 1A); group C was the coprophagy group in which rabbits wore a wide collar (7.0 cm) preventing them from eating soft feces (CP group, Figure 1B). The experiment lasted for 70 days, in order to measure the effects of wearing a collar on rabbits’ growth. 

Twelve 28-day-old female weaned New Zealand white rabbits with similar body weight (1.14 ± 0.12 kg) from a breeding center (western suburbs breeding filed of rabbit, Zhengzhou, China) were raised in the animal experimental center of Henan Agriculture University. The experimental room and cages were disinfected before housing rabbits. Rabbits were randomly divided into two groups (6 replicates in each group): group A was the control group and fed according to normal procedures; group B was the experimental group wearing an Elizabeth circle to prevent caecotrophy (Figure 2). The experiment lasted for 60 days, in order to measure the effects of fasting caecotrophy on rabbits’ growth and hepatic lipid metabolism.

For all rabbits, room ambient temperature was controlled at around 23 ± 1 °C and they had free access to food and water. Specifically, rabbits were fed with the same pelleted diet at the breeding farm, and the diet was replaced by the experimental diet gradually.

#### 2.1.3. Sample Collection

After a 60-day experimental period, rabbits were anesthetized and sacrificed. Blood samples were collected in a tube containing EDTA (Ethylene Diamine Tetraacetic Acid) for clinical biochemistry measurements using standard protocols (JianCheng Bioengineering Institute, Nanjing, China). Liver, spleen, kidney, lung, heart, muscle and adipose were collected and weighed. A subsample of liver was placed in liquid nitrogen immediately and stored until analysis. Body weight and feed intake were recorded every week for the two feeding experiments. These data were used to analyze changes in various growth indices and indices of liver and kidney function. 

#### 2.1.4. Histology

Liver tissues from twelve rabbits were fixed with paraformaldehyde, embedded in paraffin, and then cut into 5 µm sections. These sections were subjected to oil red O staining to measure lipid droplet accumulation under a Nikon microscope (Nikon Eclipse Ti-SR, Tokyo, Japan), and then quantified by Image-Pro Plus 6.0 software (Media Cybernetics, Houston, TX, USA).

#### 2.1.5. Serum Biochemical Indices

Serum biochemical indices were analyzed according to the manufacturer’s protocols (JianCheng Bioengineering Institute, Nanjing, China). A Super microplate reader (Bio-tek ELx808TM, Winooski, VT, USA) was used to measure high-density lipoprotein cholesterol (HDL-C), low-density lipoprotein cholesterol (LDL-C), total protein (TP), total cholesterol (TC), total glyceride (TG) and albumin (ALB). 

### 2.2. Transcriptome Analysis

Total RNA was extracted from the liver with Trizol reagent (Invitrogen, Carlsbad, CA, USA) according to the manufacturer’s instructions, then re-suspended in 50 µL RNase-free water and stored at −80 °C. The quality and concentration of the total RNA was analyzed with a NanoDrop 2000 spectrophotometer (Thermo scientific, Waltham, MA, USA). The concentration of total RNA was measured using the NanoDrop 2000 (Thermo scientific). RNA integrity was assessed using the Nano 6000 LabChip kit (Agilent technologies, CA, USA). All samples had an RNA integrity number (RIN) > 8.6.

#### 2.2.1. Liver Transcriptome Library Construction and Illumina Sequencing

High quality total RNA from rabbit liver (n = 3 per group) was used for preparation of RNA-seq libraries. After the total RNA sample was tested using the NEB Next UltraTM RNA Library Prep Kit for Illumina (Illumina, San Diego, CA, USA), it was inserted into the cDNA library after enrichment, fragmentation, reverse transcription, and terminal ligation of the mRNA. The average size of the products was 300 bp and was used for first strand cDNA synthesis followed by second strand cDNA synthesis. The double-stranded cDNA ends were then paired, tailed and ligated to the PE (Prep Enzyme) Adaptor Oligo Mix (Illumina, San Diego, CA, USA). Subsequent sequencing was performed using an Illumina HiSeqTM 2500 with a read length of pair-ended 150 bp.

The data obtained by sequencing (raw reads) was used for quality control (QC), to determine whether sequencing data were suitable for subsequent analysis. The clean reads were aligned with the rabbit genome by removing the linker sequence, removing reads containing >5% fuzzy reads, and removing low quality reads. Pre-experiments were performed to optimize the alignment parameters. In addition, a clean reading of greater than 30% (Q30) of the mass value in each sample was calculated (Appendix A). The clean readings were mapped to the rabbit reference genome using TopHat2 software with >80% alignment efficiency (Appendix A). 

#### 2.2.2. Data Analysis

Based on TopHat2 (http://ccb.jhu.edu/software/tophat/index.shtml) alignment of Reads and reference sequences of each sample, the Cufflinks software (Cufflinks, Berkeley, CA, USA) was used for splicing and expression quantification. By comparison with known genome annotation files, a new transcriptome region, the new genes were identified. These were aligned with each database by BLAST to obtain annotation information. The quantification of gene expression levels was estimated via fragments per kilobase of million fragments mapped. Differential expression analysis (*DEG*) was performed using the DESeq R package, which provides statistical routines for determining differential expression using a model based on the negative binomial distribution. The resulting P value was adjusted using Benjaming and Hochberg’s multiple testing correction for controlling false discovery rate (FDR). Here, only unique reads with an absolute value of log2 ratio ≥ 1 and FDR significance score < 0.01 were used for subsequent analysis. 

Gene function was annotated based on the following databases: Nr (NCBI non-redundant protein sequence (ftp://ftp.ncbi.nih.gov/blast/db/); Nt (NCBI non-redundant nucleotide sequence); Pfam (Protein family http://pfam.xfam.org/); KOG/COG (Clusters of Orthologous Groups of proteins) http://www.ncbi.nlm.nih.gov/KOG/; Swiss-prot (A manually annotated and reviewed sequence database http://www.uniprot.org/); KO (KEGG Ortholog database http://ccb.jhu.edu/software/tophat/index.shtml); GO (gene Ontology http://www.geneontology.org/). 

### 2.3. RT-qPCR

Real-time quantitative PCR (RT-qPCR) was used to measure the expression of 9 differentially expressed genes to validate our Illumina sequencing data. Total RNA was extracted from livers of rabbits in the control and experimental groups (with six replicates in each group). Primers were designed based on the assembled transcriptome sequence using primer 6 software (Primier Biosoft, Palo Alto, CA, USA) and are listed in the Appendix A. The first strand cDNA was synthesized from 1 μg of RNA by using PrimeScript^TM^ RT reagent Kit with gDNA Eraser (TaKaRa, China, Dalian). All cDNA products were diluted to 200 ng/μL. The 20 μL qRT-PCR reaction mixture consisted of 2 μL template cDNA, 0.4 μL of each primer, 10 μL of TB Green^TM^ Premix Ex Taq^TM^ (2X), 0.4 μL ROX and 6.8 μL of nuclease-free water. PCR amplification was performed in a 96-well optical plate at 95 °C for 2 min, followed by 40 cycles of 95 °C for 30 s, 60 °C for 30 s, and a final extension at 72 °C for 2 min. qRT-PCR was performed using the StepOne plus real-Time PCR system (Applied Biosystems) and 2^−ΔΔCT^ method was used to calculate the relative expression level of each gene [36]. β-actin was used as the internal control gene for normalization. 

### 2.4. Data Analysis

Data were analyzed with SPSS statistics 22 (SPSS, Chicago, IL, USA), statistical differences were determined with an unpaired two-tailed analysis (T-test). Comparisons with *p* ≤ 0.05 were considered statistically significant. Results are expressed as means ± SD.

The Pearson correlation coefficient between the fold changes calculated with the RNA-Seq and qRT-PCR data and phenotypic (including serum biochemical indices and internal organ weight) and *DEG* data were determined by SPSS 22.0 using a one-way ANOVA followed by Duncan’s multiple range test. Differences were accepted as statistically significance when *p* < 0.05. *p*-values were corrected using FDR.

## 3. Results

### 3.1. Effects of Wearing a Collar on Rabbit Growth

Compared with the control group (CON group), wearing a collar but not fasting caecotrophy (CA group) had no significant effect on total feed intake, average daily feed intake, average daily body weight gain and the final body weight (*p* > 0.05). In contrast, average daily body weight gain and final body weight in rabbits wearing a collar and performing fasting caecotrophy (CP group) were significantly decreased compared with the other two groups (*p* < 0.05, Table 1).

### 3.2. Body Weight and Liver Histology Changes

As shown in Table 2, compared with the control group, body weight, food intake, specific growth rate (SGR) and feed conversion rate (FCR) in the experimental group of rabbits were significantly decreased (*p* < 0.05), while the hepatosomatic index (HSI) did not differ (*p* > 0.05). During the experimental period, food intake of these two groups did not differ (Figure 3).

After slaughter, organs were used to measure the effects of fasting caecotrophy on development. As shown in Figure 4, compared with the control group, fasting caecotrophy resulted in a significant decrease of liver weight and perirenal fat weight (*p* < 0.05). No effect was detected on the weight of other internal organs (lung, kidney, heart and spleen). In order to measure the effects of fasting caecotrophy on lipid droplet accumulation, liver tissue was used for Oil red O staining. The Oil red O staining results, together with the quantification data, indicated that lipid droplet accumulation in the liver tissue of the experimental group was significantly lower than the control group (Figure 5, *p* < 0.05).

### 3.3. Serum Biochemistry

As shown in Figure 6, the high-density lipoprotein cholesterol (HDL-C), low-density lipoprotein cholesterol (LDL-C), total cholesterol (TC), triglycerides (TG), total protein (TP) and albumin (ALB) did not differ between the control and experimental group (*p* < 0.05). 

### 3.4. Liver Transcriptomic Analysis 

A total of 14 964 genes were expressed in liver tissue of the 90-day-old New Zealand white rabbits. In all, 444 *DEG* were detected between experimental and control group, 262 of which were down-regulated and 182 were up-regulated in the experimental group (Figure 7 and Appendix A). The *DEG* were enriched with 63 GO terms (Appendix A) and were mapped onto 217 KEGG pathways (Appendix A). Most *DEG* were enriched in lipid metabolism-related pathways, and 5% of *DEG* were enriched in the retinol metabolic pathway (Figure 8).

In order to validate the reliability of transcriptome data, we performed RT-qPCR verification of 9 randomly selected genes (including *ABCA1*, *ABCB1*, *ABCB4*, *ABCG1*, *ACACA*, *ACACB*, *CYP7A1*, *SREBP1* and *GPAM*) and compared the results with transcriptome sequencing data. Pearson correlation analysis suggested that RT-qPCR results were consistent with the transcriptome sequencing results (Pearson’r = 0.929, Figure 9), underscoring the reliability of the transcriptome data. 

Correlation analysis of 8 *DEGs* related to lipid metabolism and phenotypic data including serum biochemical indices and internal organs. As shown in Figure 10, significant correlations were not observed in the present study.

To further validate the effects of fasting caecotrophy on retinol metabolism, we performed qRT-PCR to measure the mRNA expression changes of 9 retinol metabolism-related genes in the liver of both control and fasting caecotrophy groups. Data indicated that fasting caecotrophy decreased *ADH4* and increased *CYP3A1*, respectively (Figure 11, *p* < 0.01). Other genes were not affected (*p* > 0.05). Results of *ADH4* and other genes agree with the transcriptome sequencing data, while the result of *CYP3A1* is inconsistent with the RNAseq data.

## 4. Discussion

In the present study, growth performance results showed that preventing rabbits from eating their soft feces decreased the body weight, weight gain and SGR significantly. After slaughter, liver weight and perirenal fat weight of the fasting coprophagy group were lower than the control group. Further analysis was performed to measure the lipid droplet accumulation in liver tissue between these two groups, oil red O staining results of the liver tissue indicated that lipid droplet accumulation was decreased in the experimental group compared with the control group. The quantified data of lipid droplet accumulation further confirmed the staining data, which is consistent with a previous study [35]. Under normal dietary conditions, preventing rabbits from eating soft feces led to malnutrition [37,38]. It is speculated that this negative effect occurs for two main reasons: the first one is that soft feces are rich in vitamins and essential amino acids in microbial cells, thus, at the same feed intake, rabbits that were prevented from eating feces would absorb 15% to 22% less amino acids; the second reason is that soft feces contain a large number of microorganisms, thus, preventing the rabbits from eating soft feces will result in changes in the microflora harboring the digestive tract with a consequent reduction in bacterial populations [39,40]. 

However, rabbits eating soft feces can obtain 83% of niacin, 100% of riboflavin, 165% of pantothenic acid, and 42% of vitamin B12 requirements daily, which greatly improve their utilization of sodium and potassium [37,41,42]. Studies on rats showed that eating soft feces inhibits growth [43]. R Franz found that coprophagy was an important factor explaining the ringtail possums’ low requirement for nitrogen and its ability to subsist solely on a diet of Eucalyptus foliage [44]. Experiments with guinea pigs showed that nutrient digestion rate is highly-dependent on eating feces, with soft feces eaten by the animal containing more nitrogen (and presumably amino acids) than discarded feces [45,46]. The amount of methionine and lysine obtained from soft feces by adult beavers accounted for 26% and 19% of their total intake, respectively [47,48]. Altogether, these studies underscore that nutrients in feces are critical for the growth and development of small mammals. 

By using cloning and sequencing techniques, D. H. sánchez explored the profiles of main bacteria in rabbits’ caecum and found that 94% of the bacteroides consist of firmicutes, a small amount of micro-bacteria and proteobacteria [49]. By using CE-SSCP (capillary electrophoresis single-strand conformation polymorphism) and qPCR techniques, Sylvies Combes measured the establishment of cecal microbiota in 2-day-old and 70-day-old rabbits and found that the caecum flora evolved from a simple, unstable community to a complex, top-level community [50]. The number of muramidomycetes reached a maximum, with the total number of bacteria and bacteroides-populus reaching the highest values at 21 days of age [51]. In contrast, the proportion of propionic-acid-generating bacteria decreased with age. The intestinal microflora and the host are mutually beneficial and play an important role in controlling physiological functions including the host’s metabolism, immune system development, and disease resistance [52,53]. The importance of the intestinal flora has led to the proposal that it represents a “super organ”, which is necessary for survival in humans and animals [54]. 

Although many studies have investigated the effects of eating soft feces on rabbits’ growth and physiological outcomes, they have been focused on phenotypic data and serum biochemical indicators. Some of these studies indicated that the body weight of rabbits in the CP group was lower than the control group; in the present study, the weight gain of male rabbits was higher than the female rabbits in the fasting caecotrophy group compared with the control group. However, there is no study aimed to investigate the regulatory mechanisms underlying this phenomenon. In the present study, the KEGG annotation indicated that differentially expressed genes are mainly associated with retinol metabolism, pentose and glucuronide interactions, starch and sucrose metabolism, fatty acid degradation, steroid hormone biosynthesis. These results suggested that consuming feces brought about changes in lipid metabolism in New Zealand white rabbits, and we speculate that lipid metabolism changes were one of the main reasons mediating the effects of fasting caecotrophy. 

The liver is one of the most active organs associated with lipid metabolism, and it also stores more than 90% of the retinol in the body [55]. Retinol is important in participating in the transport of plasma from the liver to peripheral organs, and this process requires the participation of retinol-binding proteins (RBP) [56]. Studies have shown that *RBP4* protein levels are positively correlated with TC and negatively correlated with HDL [57]. It is speculated that *RBP4* enhances insulin resistance by regulating hepatic TC synthesis and VLDL (Very Low Density Lipoprotein) release into blood, and finally affects fatty acid metabolism [58]. In the present study, there was no significant difference in the mRNA expression level of *RBP4* between the two groups, and the difference of TC and HDL were not significant, which implied that preventing intake of soft feces did not affect the expression of the *RBP4* gene in rabbits. 

Studies have shown that members of the cytochrome *P450* family and *ADH* are involved in the process of retinol metabolism [59]. The *CYP1* family catalyzes the conversion of retinol into retinal—a step that requires the participation of alcohol dehydrogenase (ADH)—and is ultimately converted into retinoic acid [60]. Knocking out of *CYP1* family receptor down-regulated the expression of retinoic acid [61]. In the present study, the expression of *CYP1A1*, *CYP1A2*, and *ADH4* genes in the experimental group was down-regulated compared with the control group, indicating that preventing the intake of feces had a great effect on the metabolism of retinol. 

The enzyme *CYP26* is key for regulating the expression of retinoic acid, degrading retinoic acid to an inactive product that is eventually excreted. Studies have shown that the injection of *CYP26A1* mRNA into Xenopus embryos led to symptoms of retinoic acid deficiency, suggesting that *CYP26A1* is involved in the regulation of retinoic acid in vivo and the maintenance of normal embryo development [62]. Using the zebrafish embryo as a testing material, results showed that the overexpression of *CYP26C1* brought about similar symptoms as the overexpression of *CYP26A1* and the addition of retinal dehydrogenase inhibitors during the embryonic period, indicating that *CYP26C1* is also involved in the process of folic acid metabolism [63]. 

In the present study, mRNA expression of *CYP26* was down-regulated in the experimental group compared with the control group, indicating that fasting caecotrophy may lead to the loss of retinoic acid. Studies have shown that retinoic acid plays an irreplaceable role in embryonic development, organ formation, and the proliferation and apoptosis of cells [64]. Its lack or excess can cause abnormal organ development or embryo death, thus, maintaining the balance of retinoic acid metabolism in the body is of great importance. However, the present data underscores the fact that it should not be regulated by diet alone. Attention should also be paid to the activity of enzymes that synthesize and metabolize retinoic acid, including *RDH*, *RALDH* and *CYP26* [65]. 

Under the present experimental conditions, wearing a collar was effective in preventing rabbits from eating soft feces. However, there is some skepticism among researchers that wearing a collar will influence rabbits comfort and growth. In order to assess the effects caused by wearing a collar, we performed another 70-day long feeding experiment with a narrow collar not prevent caecotrophy and a wide collar prevent caecotrophy. Results indicated that wearing this collar had no effect on rabbit growth. Although this additional experiment was performed on Angora long-haired rabbits, we believe that results are also pertinent to New Zealand rabbits. Although the correlation analysis of phenotypic data and the 8 *DEGs* did not reach statistical significance, the positive correlations suggest that using a greater number of biological replicates would lead to statistically significant results. As such, more research is needed to better ascertain the underlying regulatory mechanisms.

## 5. Conclusions

In summary, our data indicated that fasting caecotrophy decreases the body weight, liver weight and perirenal fat content in rabbits, reduces the specific growth rate and feed conversion ratio, lowers accumulation of lipid in liver. Thus, lipid synthesis may be decreased by fasting caecotrophy. Transcriptomic sequencing analysis of the liver tissue revealed a total of 301.2 million raw reads approximately 45.06 Gb of high-quality clean data. A total of 444 *DEGs* were identified with many being enriched in lipid metabolism. Further analysis revealed positive correlations of lipid metabolism genes and serum biochemical indices, liver weight and perirenal fat content. In conclusion, results from the present study provide strong evidence that reduced growth rate and lipid droplet accumulation caused by fasting caecotrophy may be mediated by impaired lipid synthesis. As such, results help better understand the effects of soft feces’ intake on the growth and development of New Zealand white rabbits and potentially of other small mammals.

## Figures and Tables

**Figure 1 animals-09-00648-f001:**
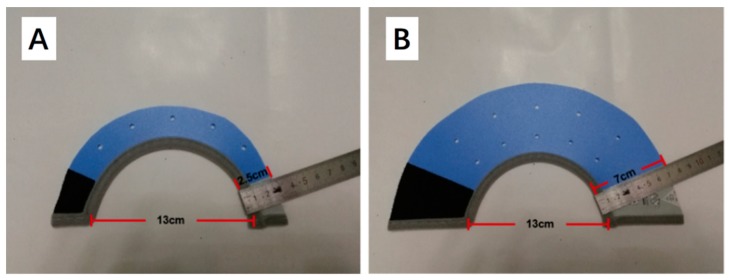
Narrow collar and wide collar used for rabbits. (**A**) Narrow collar (2.5 cm wide); (**B**) Wide collar (7.0 cm wide).

**Figure 2 animals-09-00648-f002:**
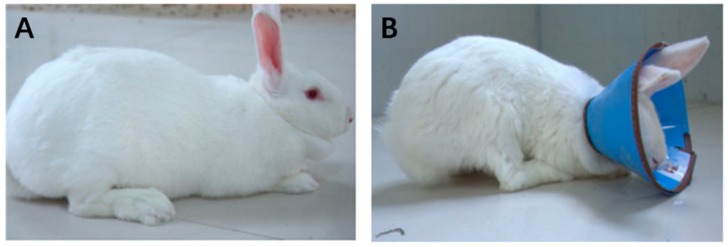
Rabbits in the two groups. (**A**) Control group; (**B**) Experimental group.

**Figure 3 animals-09-00648-f003:**
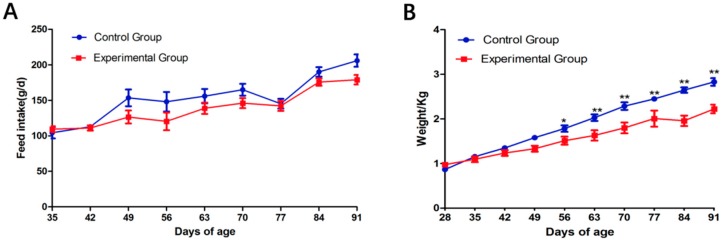
Changes in feed intake (**A**) and body weight (**B**) between the control and experimental group. Values are presented as means ± SD. *, *p* < 0.05; **, *p* < 0.01.

**Figure 4 animals-09-00648-f004:**
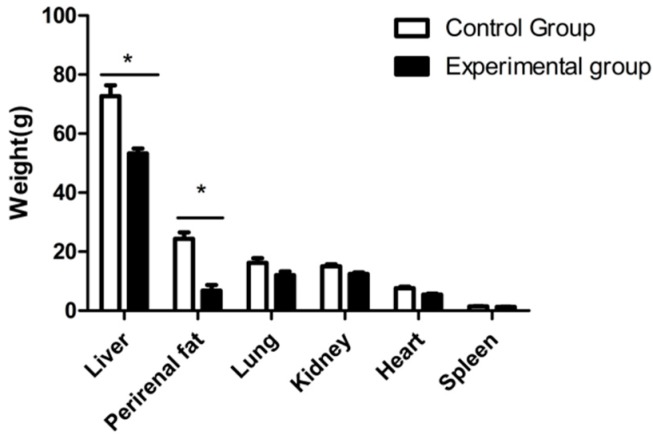
Weight changes of internal organs between the control and experimental groups. Values are presented as means ± SD. *, *p* < 0.05.

**Figure 5 animals-09-00648-f005:**
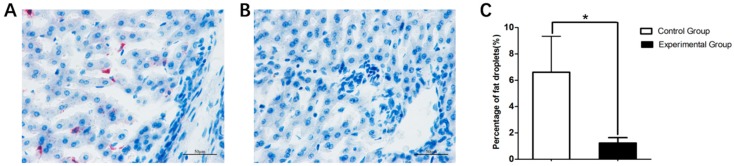
Fasting caecotrophy decreased liver lipid droplet accumulation. Lipid droplets were stained with Oil red O ((**A**): control group; (**B**): experimental group. Scale bar = 50 μm; (**C**): % area covered with Oil red O staining as measured by using Image-Pro Plus 6.0). (**A**,**B**) are representative images of 3 biological replicates. Values are presented as means ± SD. *, *p* < 0.05.

**Figure 6 animals-09-00648-f006:**
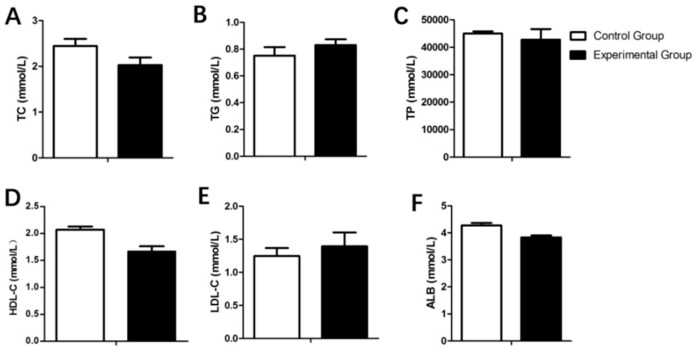
Effects of fasting caecotrophy on serum biochemical indices ((**A**): total cholesterol (TC); (**B**): triglycerides (TG); (**C**): total protein (TP); (**D**): high-density lipoprotein cholesterol (HDL-C); (**E**): low-density lipoprotein cholesterol (LDL-C); (**F**): albumin (ALB)) in rabbits. Values are presented as means ± SD.

**Figure 7 animals-09-00648-f007:**
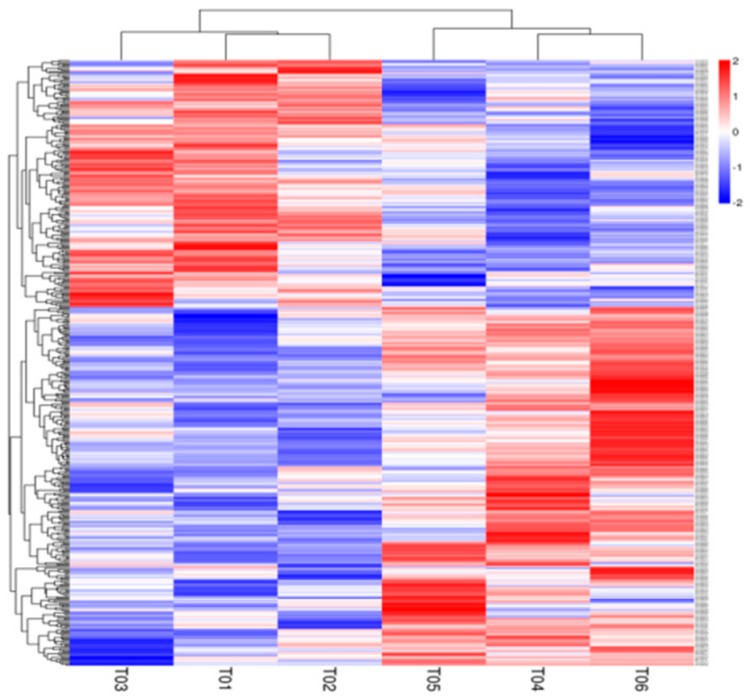
Characteristics of mRNA expression levels between the experimental and the control group. Hierarchical clustering map of differentially expressed genes. The abscissa represents the sample name and the clustering result of the sample, and the ordinate represents the clustering result of the differential gene and the gene. The different columns in the figure represent different samples, with different rows representing different genes. The color represents the level of expression of the gene in the sample log10. T01, T02, T03 were representative of the control group. T04, T05, T06 were representative of the experimental group, zero-mean normalization (z-score) was used to process the data.

**Figure 8 animals-09-00648-f008:**
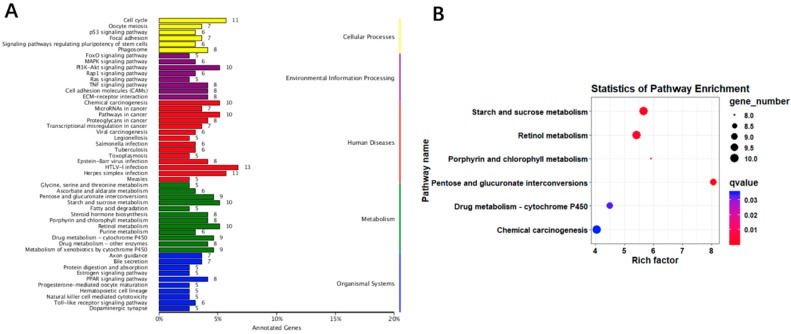
Prediction and functional analysis of target genes regulated by differentially expressed mRNAs between the control and experimental group. (**A**) KEGG classification map of the differentially expressed genes. (**B**) Scatter plot of KEGG pathways enriched in the differentially expressed genes. In Figure 8A, the ordinate represents the name of the KEGG metabolic pathway, and the ordinate the ratio of the number of genes annotated to the pathway along with the number of genes in the annotation. In Figure 8B, each circle in the figure represents a KEGG pathway, the ordinate represents the pathway name, the abscissa is the Enrichment Factor, which represents the ratio of the proportion of genes annotated to a pathway in a differential gene to the proportion of genes in all genes annotated to that pathway. The larger the enrichment factor, the more significant the level of enrichment of differentially expressed genes in this pathway.

**Figure 9 animals-09-00648-f009:**
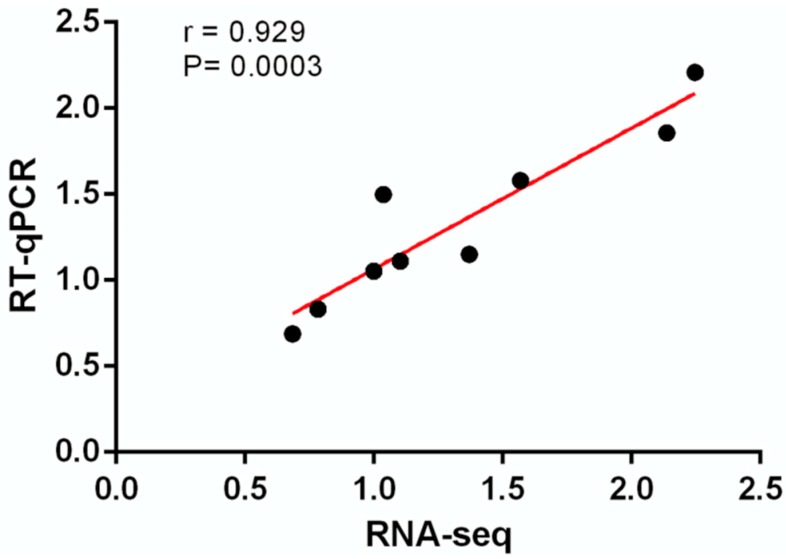
Pearson correlation analysis of quantitative real-time PCR qRT-PCR results and transcriptome sequencing data.

**Figure 10 animals-09-00648-f010:**
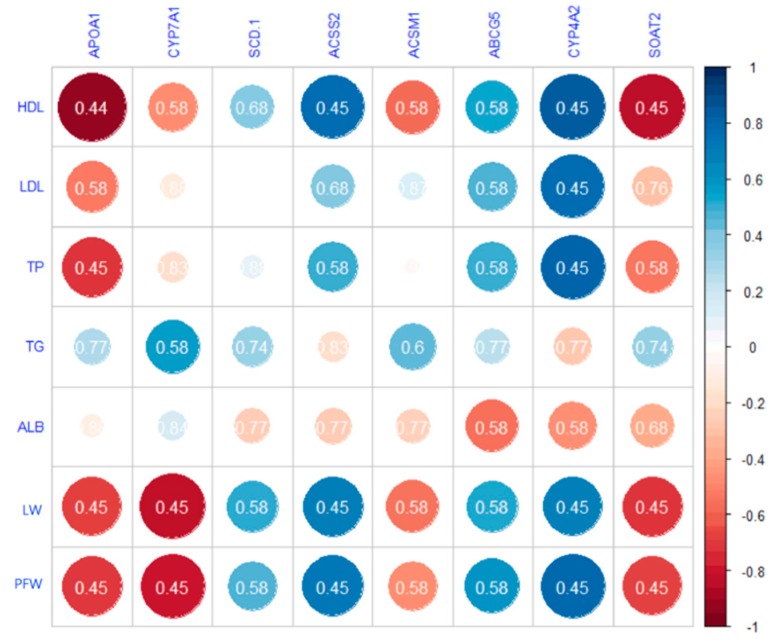
Pearson’s correlations between gene transcription and phenotypic traits. Size of the circle in the figure represents the correlation coefficient. Numbers in the circles represent *p*-values which are corrected by false discovery rate (FDR). HDL: high density lipoprotein; LDL: low density lipoprotein; TP: total protein; TG: triglycerides; ALB: albumin; LW: liver weight; PFW: perirenal fat weight.

**Figure 11 animals-09-00648-f011:**
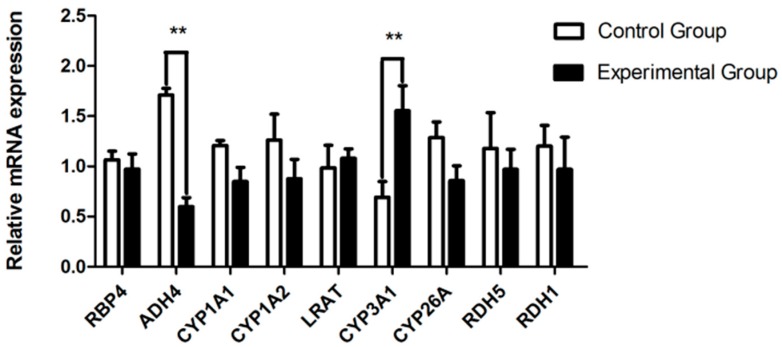
qRT-PCR validation of retinol metabolism related genes. The expression of selected genes was normalized to β-actin. *, *p* < 0.05; **, *p* < 0.01.

**Table 1 animals-09-00648-t001:** Effects of wearing a collar and fasting caecotrophy on growth performance of Angora long-haired rabbits.

Items		Groups		
		CON	CA	CP
IBW (g)		1 300.42 ± 128.02	1 288.75 ± 129.91	1 273.54 ± 116.05
ADFI (g)	Total Feed intake/g	143.98 ± 10.32	140.48 ± 10.01	138.60 ± 7.87
	Feed intake at daylight/g	48.12 ± 8.41	43.21 ± 3.99	47.29 ± 4.64
	Feed intake at night/g	95.86 ± 8.49	97.27 ± 10.93	91.40 ± 5.76
ADG (g)		21.60 ± 3.52 ^a^	21.69 ± 2.11 ^a^	19.64 ± 1.98 ^b^
Final weight (g)		2 736.36 ± 271.13 ^a^	2 722.00 ± 165.15 ^a^	2 524.76 ± 211.59 ^b^
F/G		6.80 ± 1.08	6.52 ± 0.61	7.12 ± 0.75

IBW: Initial body weight; ADFI: Average daily feed intake; ADG: Average daily body weight gain; F/G: Feed intake/ body weight ratio. Values in this table are presented as means ± SD. In the same row, values with no superscript letters denote significant differences at *p* > 0.05, while values with different superscript letters denote significant difference at *p* < 0.05.

**Table 2 animals-09-00648-t002:** Growth index changes of the control group and fasting caecotrophy group.

Items	Treatment	
	Control Group	Experimental Group
Initial weight (kg)	0.913 ± 0.118	0.922 ± 0.163
Final weight (kg)	2.935 ± 1.188 ^a^	2.292 ± 0.170 ^b^
Weight gain (%)	2.021 ± 0.159 ^a^	1.370 ± 0.262 ^b^
SGR (%day-1)	0.033 ± 0.002 ^a^	0.022 ± 0.004 ^b^
FCR	3.555 ± 0.341 ^a^	4.795 ± 1.238 ^b^
HIS (%)	44.702 ± 4.918	45.971 ± 3.971

SGR: Specific growth rate = (Final body weight − Initial body weight)/Feeding days; FCR: Feed conversion rate = (Increased body weight/Feed intake); HIS: Liver index = (Liver weight/body weight). Values in this table are presented as means ± SD. In the same row, values with no superscript letters denote significant differences at *p* > 0.05, while values with different superscript letters denote significant difference at *p* < 0.05.

## Data Availability

All raw data have been deposited to NCBI Sequence Read Archive (SRA accession: PRJNA504548, Temporary Submission ID: SUB4763199).

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
