# Peer review of "Transcriptome Analysis of the Effects of Fasting Caecotrophy on Hepatic Lipid Metabolism in New Zealand Rabbits"

_animals, 2019, doi:10.3390/ani9090648_

Round 1

Reviewer 1 Report

Wang and colleagues have provided a revised version of their study on the effects of caecotrophy on lipid metabolism in rabbits. They did perform new experiments to evaluate if wearing a collar was affecting food and water intake in their experimental set up which is critical to conclude that this device does not confound the reported differences. Although it seems to be a trend towards the collar affecting all the measured parameters, these differences did not reach statistical significance. Before considering this manuscript for publication, the authors have to address a number of issues related to the RNAseq experiments.

1- RNAseq: the DEG genes shown were selected based on a log2 ratio >1 and FDR <0.01 (lane 178-179) and in figure 11 some of those genes are not showing that behaviour, how the authors can then claim that their RNAseq data is reliable ?

2- Any of the correlations in figure 10 is significant. A p-value of ~0.4 is not supporting that any of those correlations is different to 0 (even arguing that the sample size was small!). Authors cannot draw conclusions from this analysis.

3- Figure 8B. Represented Q-values should be from 0-0.05. Based on the colour scale, it is difficult to ascertain if any of those pathways is significantly enriched in the experimental group.

4- There is a number of typos and grammatical errors. Please revise the manuscript thoroughly.

Author Response

Reviewer 1

Comments and Suggestions for Authors

Wang and colleagues have provided a revised version of their study on the effects of caecotrophy on lipid metabolism in rabbits. They did perform new experiments to evaluate if wearing a collar was affecting food and water intake in their experimental set up which is critical to conclude that this device does not confound the reported differences. Although it seems to be a trend towards the collar affecting all the measured parameters, these differences did not reach statistical significance. Before considering this manuscript for publication, the authors have to address a number of issues related to the RNAseq experiments.

1- RNAseq: the DEG genes shown were selected based on a log2 ratio >1 and FDR <0.01 (lane 178-179) and in figure 11 some of those genes are not showing that behaviour, how the authors can then claim that their RNAseq data is reliable ?

AU: Thanks for your comments. We agree with the reviewer’s comments that RNAseq data should be consistent with the RT-qPCR results. In the present study, we performed RT-qPCR experiment to validate the results of RNAseq data, our results of Pearson correlation analysis suggested that RT-qPCR results were consistent with the RNAseq data (Pearson’r=0.929), please see them in figure 9 and the description are presented in Line 295-300. For figure 11, we are sorry for that we did not make it clear in the description, in fact, this experiment was designed because we want to verify some important genes related to retinol metabolism (liver stores more than 90% of the retinol in the body ). For these genes, only ADH4 was the DEG gene identified by RNAseq data, the other genes are not DEG genes. From figure 11, we can see that CYP3A1 was significantly increased in the experimental group compared with the control group, while CYP3A1 was not DEG gene in the RNAseq data, so results of this gene are inconsistent between RT-qPCR and RNAseq data (this may be caused by some errors), the other genes are all the same. We have made revisions in the manuscript, please see them in Line 320-321.

2- Any of the correlations in figure 10 is significant. A p-value of ~0.4 is not supporting that any of those correlations is different to 0 (even arguing that the sample size was small!). Authors cannot draw conclusions from this analysis.

AU: Thanks for your comments. We have rewrote the description of figure 10 as suggested, please see them in Line 309-310.

3- Figure 8B. Represented Q-values should be from 0-0.05. Based on the colour scale, it is difficult to ascertain if any of those pathways is significantly enriched in the experimental group.

AU: Thanks for your comments, we agree with you that the Q-values should be from 0-0.05. So we have remade figure 8B and presented pathways that are significantly enriched in the experimental group. Please see them in figure 8B.

4- There is a number of typos and grammatical errors. Please revise the manuscript thoroughly.

AU: We are sorry for the typos and grammatical errors. Now we have revised the manuscript thoroughly, please see them in the manuscript.

Reviewer 2 Report

Manuscript has been improved significantly. I found some minor errors:

line 81 I think that authors wrongly interpreted the results of the study By Zang et al (position 20 of refernces)

Fig 10 - Please specify if the numbers in circles are p- values or FDR, please replace "liver" with "liver weight" and "perirenal fat" with "perirenal fat weight" on the figure 10 o0r use some short cuts

Author Response

Comments and Suggestions for Authors

Manuscript has been improved significantly. I found some minor errors:

line 81 I think that authors wrongly interpreted the results of the study By Zang et al (position 20 of refernces)

AU: We are sorry for the improper citation. A new reference have been added in the manuscript to support the description. Please check them in the manuscript.

Fig 10 - Please specify if the numbers in circles are p- values or FDR, please replace "liver" with "liver weight" and "perirenal fat" with "perirenal fat weight" on the figure 10 o0r use some short cuts

AU: In figure 10, numbers in the circles represent p-values, we have revised the figure note to make it clear. And we have replaced “liver” with “liver weight” and “perirenal fat” with “perirenal fat weight” in figure 10. Please see them in the manuscript.

Round 2

Reviewer 1 Report

Authors have addressed all my comments. However, I can see that the colour legend in figure 8 still shows qvalues from 0-1. Please modify.

Author Response

Comments and suggestions:

Authors have addressed all my comments. However, I can see that the color legend in figure 8 still shows qvalues from 0-1. Please modify.

AU: Thanks for your comments. We have modified figure 8 and changed qvalues into 0-0.05 as suggested, please check it in the manuscript.

This manuscript is a resubmission of an earlier submission. The following is a list of the peer review reports and author responses from that submission.

Round 1

Reviewer 1 Report

Wang and colleagues evaluated the importance and significance of caecotrophy habits in rabbits in regulating metabolism. Overall this paper is poorly drafted and the manuscript lacks of cohesion and coherence, and rigorousity in the experiment description and justification. The results are presented as chunks in the text without justifying why the analyses were done. Because of this, the paper is pretty hard to follow and this referee could not properly evaluate if the experiments were rigorously performed. Apart from this, this manuscript has to be extensively revised by a native speaker to address the large number of English grammar mistakes.

Additional comments

1) Methods (e.g. 2exp-deltaCT), software (e.g.TopHat), databases (e.g. KEGG) (for citing some) do have to be referenced in the manuscript.

2) Method section is poorly described and I am not sure if the authors know what they have done in the RNAseq experiments. What type of libraries did they generate (paired end, single end), length of fragments, chemistry?, what was the average size of the libraries (not total)? ,how redundant seqences were eliminated?, is there a reference genome? if yes, this has to be referenced in the text. If yes, this is not a de novo assembly and therefore this does not require annotation.

3)Which protocol was used for the malate dehydrogenase assay?

4)how the feed conversion rate was calculated?

5)What does the figure 1-5 represent? Mean+- SD?

6)Volcano plot is useless

7)Is figure 6B representing fold change?

8) In figure 7B the enrichement factor (size of the circles) represents the proportion of differencial expressed genes found by the authors in their experiments over the total number of genes assigned to that specific Pathway. It is not clear at all the description of this parameter given by the authors.

9)What is the rationale for including those specific genes and metabolites (and not others) in the pearson correlation in figure 9? Were the p-values corrected for multiple comparisons?

10) In the  introduction and discussion, the authors refer to the gut microbiota, but no experiments are provided on this. What is the relevance of the microbiota to this manuscript?

11)RNAseq data has to be deposited in a public repositorie (e. g. SRA)

Author Response

Wang and colleagues evaluated the importance and significance of caecotrophy habits in rabbits in regulating metabolism. Overall this paper is poorly drafted and the manuscript lacks of cohesion and coherence, and rigorousity in the experiment description and justification. The results are presented as chunks in the text without justifying why the analyses were done. Because of this, the paper is pretty hard to follow and this referee could not properly evaluate if the experiments were rigorously performed. Apart from this, this manuscript has to be extensively revised by a native speaker to address the large number of English grammar mistakes.

AU: Thanks for your comments, we are sorry for all the issues, we have checked thoroughly throughout the manuscript, rewrite the introduction and conclusion, and the manuscript have been revised by a native speaker to address the grammar mistakes. Please see them in the manuscript.

Additional comments

1) Methods (e.g. 2exp-deltaCT), software (e.g.TopHat), databases (e.g. KEGG) (for citing some) do have to be referenced in the manuscript.

AU: Thanks for your suggestion, we have added references for 2-ΔΔCT method and websites for software and databases in the manuscript as suggested.

2) Method section is poorly described and I am not sure if the authors know what they have done in the RNAseq experiments. What type of libraries did they generate (paired end, single end), length of fragments, chemistry?, what was the average size of the libraries (not total)? ,how redundant seqences were eliminated?, is there a reference genome? if yes, this has to be referenced in the text. If yes, this is not a de novo assembly and therefore this does not require annotation.

AUWe are sorry for our poorly description, and thanks very much for your question. We have rewrite this part in the manuscript as follows: High quality total RNA from rabbit liver (n=3 per group) was sent to prepare an RNA-seq library. After the total RNA sample was tested using the NEB Next UltraTM RNA Library Prep Kit for Illumina (NEB, USA) , the cDNA library was inserted into the cDNA library after enrichment, fragmentation, reverse transcription, and terminal ligation of the mRNA. The average size is 300 bp. This was used for first strand cDNA synthesis followed by second strand cDNA synthesis. The double-stranded cDNA ends were then paired, tailed and ligated to PE Adaptor Oligo Mix (Illumina). Subsequent sequencing was performed using an Illumina HiSeqTM 2500 with a read length of double-ended 150 bp.

The data obtained by sequencing is called raw reads, and then quality reads (QC) are performed on the raw reads to determine whether the sequencing data is suitable for subsequent analysis. The clean reads were aligned with the rabbit genome by removing the linker sequence, removing reads containing >5% fuzzy reads, and removing low quality reads. Pre-experiments were performed to optimize the alignment parameters. In addition, a clean reading of greater than 30% (Q30) of the mass value in each sample was calculated (S1 map). The clean readings were mapped to the rabbit reference genome using TopHat2 software with > 80% alignment efficiency (S2 map).

3)Which protocol was used for the malate dehydrogenase assay?

AU: We are sorry for the mistake, we have deleted the malate dehydrogenase assay in the manuscript. See them in line .

4)how the feed conversion rate was calculated?

AU: SGR:Specific growth rate=(Terminal body weight-Initial body weight)/Feeding days

    FCR:Feed conversion rate=(Increased body weight/ Feed intake)

HIS:Liver index=(Liver weight/body weight)

We have added the calculation method of feed conversion rate in the manuscript under Table 1, the full name and calculation method SGR and HIS were also added in the manuscript.

5)What does the figure 1-5 represent? Mean+- SD?

AU: Thanks for your comments. In this manuscript, all the data are presented as means±SD, we have added description in the manuscript, see them in figure 2-5 and table 1.

6)Volcano plot is useless

AUThanks for your comments, we have deleted the volcano plot as suggested.

7)Is figure 6B representing fold change?

AUThanks for your question. Figure 6B does not represent the foldchange value, it represents expression level which adjusted by z-score transformation for each DEG gene.

Zero-mean normalization is also called standard deviation normalization. The processed data has a mean of 0 and a standard deviation of 1. The conversion formula is: x* = (x - μ ) / σ.  

μ is the mean of all sample data and σ is the standard deviation of all sample data.

Since we have deleted Figure 6A as suggested, so “Figure 6B” have changed into “Figure 6”see this in the manuscript.

8)In figure 7B the enrichement factor (size of the circles) represents the proportion of differencial expressed genes found by the authors in their experiments over the total number of genes assigned to that specific Pathway. It is not clear at all the description of this parameter given by the authors.

AUWe are sorry for the unclearness. In the present study, the abscissa is the Enrichment Factor, which represents the ratio of the proportion of genes annotated to a pathway in a differential gene to the proportion of genes in all genes                                                                                                                                                                                                                                                                                                                                                                                                                                                                                                                                                                                                                                                                                                                                                                                                                                                                                                                                                                                                                                                                                                                                                                                                                                                                                                                                                                                                                                                                                                                                                                                                                                                                                                                                                                                                                                                                                                                                                                                                                                                                                                                                                                                                                                                                                                                                                                                                                                                                                                                                                                                                                                                                                                                                                                                                                                                                                                                                                                                                                                                                                                                                                                                                                                                                                                                                                                                                                                                                                                                                                                                                                              annotated to that pathway. The larger the enrichment factor, the more significant the level of enrichment of differentially expressed genes in this pathway.

9)What is the rationale for including those specific genes and metabolites (and not others) in the pearson correlation in figure 9? Were the p-values corrected for multiple comparisons?

AUThanks for your comments. In the present study, we aimed to measure the effects of fasting caecotrophy on rabbits growth and development, our in vivo data suggested that fasting caecotrophy resulted in a significant decrease of body weight and perirenal fat, so we speculated that lipid metabolism may be involved during the fasting caecotrophy period. Then we subjected the livers of the control group and fasting caecotrophy group for transcriptomic sequencing and identified many differentially expressed gens(DEGs), most of these DEGs are enriched in pathways of lipid metabolism. We also measured the changes of serum biochemical indices, there was only a mild change but not significant change between the control group and the fasting caecotrophy group, these serum biochemical indices can reflect the condition of lipid metabolism. So we did the person correlation analysis of these DEGs (lipid metabolism related genes) and serum biochemical indices, liver weight and perirenal fat, aim to elucidate whether the changes of these metabolites are caused by the changes of these DEGs genes. For the person correlation analysis, we made multiple comparisons of the data without correcting the p-value. Significant correlation was determined by primary p-value, since few samples used in the analysis.

10)In the introduction and discussion, the authors refer to the gut microbiota, but no experiments are provided on this. What is the relevance of the microbiota to this manuscript?

AUThanks for your comments. Since the soft feces of rabbits contain many nutrient substances, microorganism and bacterial, so eating soft feces will replenish them to the rabbits intestine, this is of great importance for rabbits to maintain their normal digestive function to digest high fiber content feed. Many research have found that fasting caecotrophy will affect the intestinal microbial populations of rabbits, and our team have done some research to measure the effects of fasting caecotrophy on the caecum microbial populations in rabbits, our results suggested changes in microbial population and the differentiated microbial population are mainly enriched in lipid metabolism (data not published). So we keep this part in the manuscript and added some description of microbiota, lipid metabolism in the present study.

11)RNAseq data has to be deposited in a public repositorie (e. g. SRA)

AUThanks for your suggestion. In the present study, all raw data were deposited to NCBI Sequence Read Archive (SRA accession: PRJNA504548, Temporary Submission ID: SUB4763199).

Reviewer 2 Report

The article deals with an interesting topic, from the point of view of rabbit breeders as well as from a scientific point of view. Unfortunately, the way in which the experiment was designed does not allow to unambiguously determine whether the observed changes in the growth and gene expression are caused exclusively by coprothrophy or by the wearing of the collar. There is no information if the animals wore this collar for the whole duration of the experiment. We can assume that it influenced much the comfort and welfare of animals. Therefore, an additional control,  to assess the effect of wearing the collar, is necessary in my opinion. More specific comments are included in the attached pdf file.

Author Response

The article deals with an interesting topic, from the point of view of rabbit breeders as well as from a scientific point of view. Unfortunately, the way in which the experiment was designed does not allow to unambiguously determine whether the observed changes in the growth and gene expression are caused exclusively by coprothrophy or by the wearing of the collar. There is no information if the animals wore this collar for the whole duration of the experiment. We can assume that it influenced much the comfort and welfare of animals. Therefore, an additional control,  to assess the effect of wearing the collar, is necessary in my opinion. More specific comments are included in the attached pdf file.

AUThank you for your valuable comments. In the present study, the control group rabbits were feed normally, and the fast caecotrophy group rabbits wearing the Elizabeth collar for the whole duration of the experiment. In order to assess the effects of wearing the collar on rabbits growth, we performed another experiment: rabbits were divided into three groups: the first group is the control group and rabbits were feed normally, the second group is the wearing collar not fasted caecotrophy group (rabbits wearing a narrow collar which do not fasted them from eating soft feces), the third group is the wearing collar fasted caecotrophy group (rabbits wearing a collar which fasted them from eating soft feces). As shown in suppl. table 1, our results suggested that, compared with the control group, treatment with wearing collar but not fasted caecotrophy has no significant effect on rabbits Total feed intake, Average daily feed intake, Average daily body weight gain and the final body weight, while the average daily body weight gain and the final body weight of the wearing collar fasted caecotrophy group were significantly decreased compared with the other two groups, indicating that wearing collar did not influence rabbits growth and the decease of body weight were caused by fasting caecotrophy.

Reviewer 3 Report

This is an interesting study regarding the benefits of caecotrophy on rabbits. The manuscript has great body of evidence, however, the writing seems rather poor. Additionally, the authors do not manage to clearly communicate their findings to the readership. Please find below some major comments that have to be addressed.

Major comments

1. Table 1. Please write in the footnote what the abbreviations SGR, FCR and HIS stand for.

2. Figure 5. Please write in the figure legend what the abbreviations HDL, LDL, TC, TG, TP and ALB stand for.

3. The first paragraphs of the introduction and the discussion do not belong to the present manuscript but they are guidelines for the authors and should be removed.

4. The authors should re-write the conclusions because due to poor writing there is no take-home message.

5. Please place each figure legend exactly under the respective figure without text between them.

6. Why did the authors choose rabbits as their experimental model? What are the practical applications of this study? Please comment in the discussion.

7. The authors should re-check the English language throughout the manuscript and make the appropriate corrections where necessary. 

Author Response

Reviewer 3

This is an interesting study regarding the benefits of caecotrophy on rabbits. The manuscript has great body of evidence, however, the writing seems rather poor. Additionally, the authors do not manage to clearly communicate their findings to the readership. Please find below some major comments that have to be addressed.

AU: Thanks for your comments, we are sorry for the poorly writing. Furthermore, the manuscript have been revised by a native speaker to address the grammar mistakes, please check in the manuscript.

Major comments

1. Table 1. Please write in the footnote what the abbreviations SGR, FCR and HIS stand for.

 AUWe have added description for the abbreviations of SGR, FCR and HIS as suggested, please see them in the footnote of Table 1.

2. Figure 5. Please write in the figure legend what the abbreviations HDL, LDL, TC, TG, TP and ALB stand for.

 AUThank you very much for your comments, we have added description for the abbreviations of HDL-C, LDL-C, TC, TP, TG and ALB as suggested, please see them in the manuscript.

3. The first paragraphs of the introduction and the discussion do not belong to the present manuscript but they are guidelines for the authors and should be removed.

 AUWe are sorry for this mistake, we have removed the first paragraph of the introduction and discussion in the manuscript as suggested.

4. The authors should re-write the conclusions because due to poor writing there is no take-home message.

 AUWe are sorry for our poorly writhing. We have re-wrote the conclusions and sent the manuscript to a native speaker for revision. See them in the manuscript.

5. Please place each figure legend exactly under the respective figure without text between them.

 AUwe are sorry for the disorders, we have revised the figure legend and placed them exactly under the respective figure as suggested. See them in the manuscript.

6. Why did the authors choose rabbits as their experimental model? What are the practical applications of this study? Please comment in the discussion.

 AU: Thanks for your valuable comments. We choose rabbits as the experimental model, because the aim of the present study is to measure the effects of fasting caecotrophy on animal growth, development and the possible biological mechanism. Since coprophagy is the instinct behavior for many small herbivore animals including rabbits, dogs, koala and elephant, etc. However, rabbits is thought to be the best model for its suitable size, high reproduction ratio, and easily to control when we treat them with a Elizabeth collar to prevent them from eating soft feces. So we choose rabbits as the experimental model in our study.

Coprophagy is of great biological importance for their growth and development, and many studies have reported that fasting caecotrophy will decrease the growth rate of rabbits, and evenly lead to fetal death during the late period of pregnancy in rabbits (we have done this research and these data have not been published). And our previous studies have also found that fasting caecotrophy will result in changes in caecum microbial population, metagenomics analysis suggested that the differentiated microbial population are mainly enriched in lipid metabolism (data not published). So it is very important for us to clarify the possible mechanism of fasting caecotrophy on rabbits growth, development and reproduction. The aim of our team work is to assess the effects of fasting caecotrophy on rabbits growth,  development and reproduction, and explore its possible mechanism, thus provide some evidence for rabbits production; Most importantly, we aim to identify some beneficial microbial population, and prepare some microecologics that can be used for baby rabbits to prevent them from diarrhea, finally the survival rate of baby rabbits was improved. We have added some description in the manuscript as suggested, see them in line….

7. The authors should re-check the English language throughout the manuscript and make the appropriate corrections where necessary. 

AUThank you for your comments. We have checked the manuscript thoroughly and sent the manuscript to a native speaker for revision, many corrections have been made. See them in the manuscript.

Round 2

Reviewer 1 Report

Authors have partially addressed the comments raised by this referee. However, there are a number of issues that have to be addressed before considering this manuscript for publication.

1- Although the authors state that the current version of the manuscript has been revised by a native speaker, I can still see many grammar mistakes (mainly in the discussion section).

2- Figure 9. Although the authors analysed a few number of samples, this is not an excuse to do not correct p-values for multiple comparisons. Testing many hypothesis is associated with a higher probability of reporting false positives, and this has to be controlled in their results.

Reviewer 2 Report

Authors adressed my question about  an additional control,  to assess the effect of wearing the collar, and stated that they performed an additional experiment , however in the main text they did not mention about it. In conclusion we can read: In the future, we plan to design a group of collars that can eat soft feces to see its effects. Why authors did not include additional experiment into the paper?

More comments:

line 140 (QC) stand for quality control

139 - pair-end is more often used than double -end

Reviewer 3 Report

The authors have fully addressed my comments.

Round 3

Reviewer 1 Report

This version of the manuscript still needs being reviewed by a native speaker.

I cannot trust the results from figure 9 unless the authors correct for multiple comparisons.

Reviewer 2 Report

In my opinion article should not be published unless additional experiment about effect of wearing collar on rabbit`s growth  is  described in the manuscript